# Operating Problems of Arc Plasma Reactors Powered by AC/DC/AC Converters

**Grzegorz Komarzyniec * and Michał Aftyka**

Faculty of Electrical Engineering and Computer Science, Lublin University of Technology, 20-618 Lublin, Poland; aftyka.michal@gmail.com

\* Correspondence: g.komarzyniec@pollub.pl

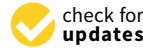

**Featured Application: Real-time control of the gliding arc plasma parameters.**

**Abstract:** The scientific objective was to investigate the cooperation of three-electrode plasma reactors with gliding arc discharge powered from multi-phase AC/DC/AC converters. In order to achieve the scientific and practical goal of the project, a test stand was designed and built, which included: a multi-electrode GlidArc type plasma reactor; a power-electronic AC/DC/AC converter, working as a source of voltage or current with regulated parameters of energy transferred to the discharge space; reactor operation diagnostics systems; and a process gas feeding and flow control system. The GlidArc Plasma Reactor has shown high sensitivity to changes in many electrical as well as gas chemical, gas-dynamic and mechanical parameters. The AC/DC/AC converter turned out to be a system sensitive to interference generated by the plasma reactor. It can be noticed that the operation of the reactor in certain conditions causes bigger interferences of the converter. However, it is difficult to systematise the influence of particular parameters of the reactor's operation on the operation of the AC/DC/AC converter and vice versa due to mutual correlations of many parameters. The correct operation of a plasma reactor depends on the characteristics of the power supply system; on the other hand, the power supply system reacts to such an untypical receiver as a plasma reactor.

**Keywords:** plasma reactor; supply system; inverter; transformer

## 1. Introduction

The power supply system is an integral part of the plasma generation system, determining the possibility of using the plasma process on an industrial scale and requiring special design and construction methods. The correct operation of a non-thermal plasma arc reactor depends on the characteristics of the power supply system [1]. On the other hand, the power supply system reacts to such an unusual receiver as a plasma reactor. A big design and operational challenge is the generation by the plasma reactor of significant electromagnetic interferences, both conductive and radiated, as well as surges [2]. The regulation of output power and its parameters within wide limits, good operating properties and high energy efficiency and resistance to interference are the most desirable features of the plasma reactor power supply system.

Generation of non-thermal plasma by means of arc discharge requires careful selection of the structure of the power supply system, i.e., topology of the system, selection of components and optimal selection of output parameters, such as values of current, voltage, their instantaneous course and frequency. Therefore, a plasma reactor power supply system is usually designed and constructed together with a plasma reactor and the required plasma process geometry, at a specific composition of process gas, its flow velocity and pressure.

Transformer power supplies work well as power supplies for plasma reactors [3]. Their disadvantage is the inability to smoothly regulate the voltage, current and frequency of the reactor power supply to compensate unwanted fluctuations in the parameters of the plasma process. A good power supply system should be able to control the parameters of the generated plasma depending on the type of process gas, its flow velocity, pressure, humidity, chemical purity and requirements of the plasma process. Such conditions can be provided by specialised power electronic AC/DC/AC converters designed to work with an arc plasma reactor.

## 2. Technical Specification of the Arc Plasma Reactor

The GlidArc plasma reactor has a modular design, consisting of two section, each section with three or six working electrodes. The sections can operate in parallel as independent plasma reactors (Figure 1B) or in cascade connection into one two-stage plasma reactor (Figure 1A). A section can have different number and geometry of electrodes, and work with different process gas as well as at other power supply parameters. This approach allows, if it is required, for each section's realising other assumptions of the plasma treatment process. The reactor design data are given in Table 1.

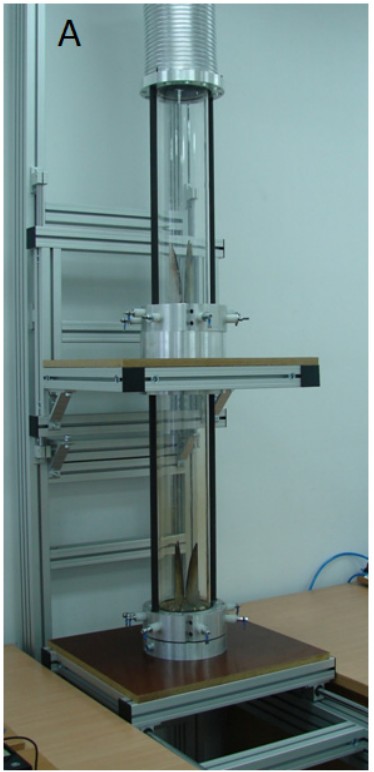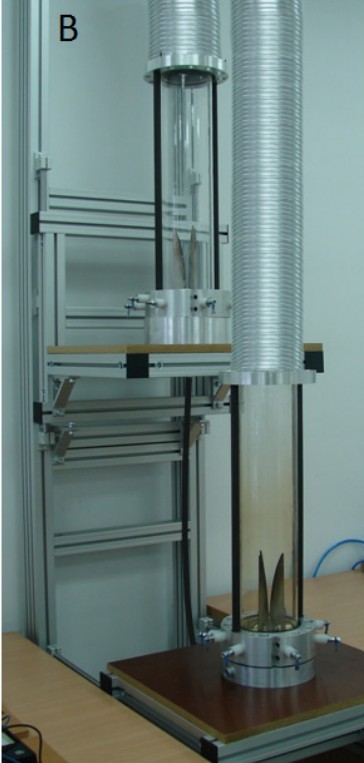

**Figure 1.** Reactor in cascade (**A**) and parallel (**B**) configuration.

The reactors' discharge chamber, with a diameter of 120 mm and a height of 500 mm, is made of a tube of heat-resistant glass, which enables visual observation of the phenomena taking place in the reactor. The working electrodes of the reactor are embedded in aluminium rings which simultaneously act as process gas inflow nozzles.

In parallel operation of reactors and for the first stage in cascade operation, the plasma gas inlet nozzle is a 5 mm diameter hole. In the second stage of the cascade, a convergent nozzle with a diameter of 112 mm is used at the gas outlet of the first stage, and up to 5 mm at the gas outlet of the second stage.

The electrodes are mounted and powered by means of metal current passes in ceramic insulation. The shape and material of the electrodes is selected according to the composition of the process gas, the plasma process, the parameters of the power supply in the process gases and electricity, the cooling conditions and the effects of the electrodes' erosion.

**Table 1.** Parameters of the plasma reactor.

| | |
|---|---|
| Number of reactor sections | 1, 2 in cascade or parallel connection |
| Height of one section | 580 mm |
| Height of the cascade | 1160 mm |
| Height of the discharge chamber of one section | 500 mm |
| Discharge chamber diameter | 114 mm |
| Discharge chamber material | quartz glass |
| Material of the electrode clamping ring | aluminium PA6 (2017A) |
| Number of working electrodes per section | 3, 6 |
| Number of ignition electrodes in one section | 1, 2 |
| Working electrode material | stainless steel 0H18N9 (or other material selected for the plasma process) |
| Ignition electrode material | tungsten, 1 mm diameter wire |
| Shape of the working electrodes | knife shape (or other shape selected for the plasma process) |
| Height of the working electrodes | 100 ÷ 250 mm |
| Electrode spacing in the ignition zone of the discharge | 2 ÷ 5 mm |
| Electrode spacing in the extinguishing zone | 30 ÷ 50 mm |
| Electrode holders | steel current passes embedded in ceramic insulators |
| Insulator material | ceramic mass AL- 70 |
| Diameter of the first stage gas inlet nozzle | 3 ÷ 8 mm |
| Diameter of the second stage gas inlet nozzle | conjugated nozzle from 112 mm to 3 ÷ 8 mm |
| Working gas flow | controlled between 0.5 ÷ 17 $m^3$/h |
| Plasma-generating gases | helium, argon, nitrogen, air and their mixtures |

As the nominal supply voltage of the working electrodes is too low to pierce the inter-electrode space, additional ignition electrodes are used in the reactor design. These electrodes facilitate the ignition of the discharge by pre-injection of the inter-electrode space. Ignition electrodes are made of short sections of tungsten wire, 1 mm in diameter, properly profiled and installed under the working electrodes.

## 3. Plasma Reactor Supply of Process Gases

When an anaerobic, inert atmosphere is needed, the plasma-generating gases are the noble gases: helium, neon, argon, krypton, xenon or radon. These elements are almost completely unreactive and the first two do not form any chemical compounds. Noble gases are too expensive to be used on a large scale in plasma generation processes, so nitrogen is the most common plasma-generating gas. If the nitrogen atmosphere proves too reactive, argon can be used as a plasma-generating gas. Due to the high content of argon in the air, it is a relatively cheap noble gas. Argon can be used as an admixture to other process gases to reduce the ignition voltage of the discharge in the reactor.

To control the atmosphere in the discharge chambers of the plasma reactor, the process gases are fed through calorimetric flow controllers (Figure 2). The process gases, selected as required from among helium, argon, nitrogen, oxygen and air, are fed through three regulators with flows of 15, 2 and 0.06 $m^3$/h. The mixing of gases takes place before the nozzle introducing them into the plasma reactor chamber.

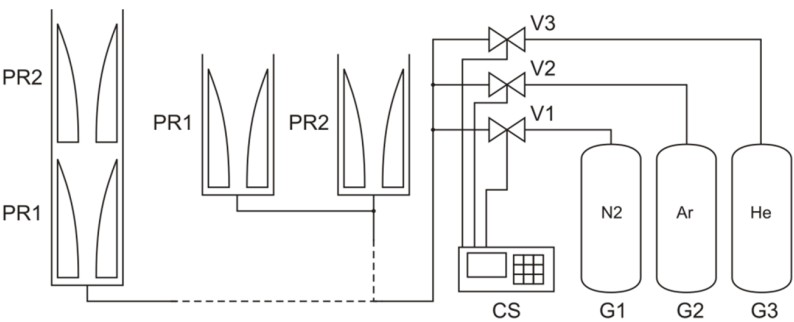

**Figure 2.** Cascade and parallel reactor connection in the process of gas supply system: PR—plasma reactor; CS—control and measurement module; V—flow regulators; G—process gas cylinders.

## 4. AC/DC/AC Power Supply

The converter power system consists of three devices: (1) AC/DC/AC converter, (2) matching transformers, (3) ignition module (Figure 3).

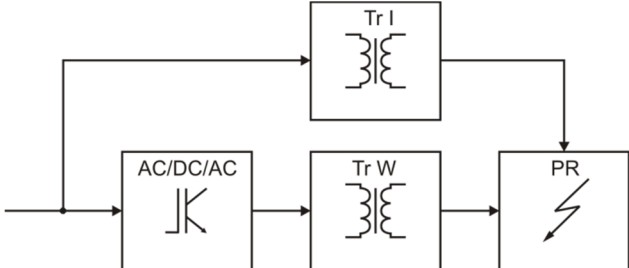

**Figure 3.** GlidArc Plasma Reactor Power Supply System: AC/DC/AC—converter, Tr W—booster transformer, Tr I—ignition system, PR—plasma reactor.

The converter power supply system of a plasma reactor consists of two converters: (1) AC/DC input (mains), (2) DC/AC output (Figure 4) [4]. The converter parameters are given in Table 2.

**Table 2.** Parameters of the power supply system.

| Operating Data | Converter | Transformer | Ignition System |
|---|---|---|---|
| Input power | 10.38 kVA | 4.6 kVA | 60 VA |
| Primary voltage | 230 V | 230 V | 230 V |
| Primary current | 15 A | 20 A | 0.25 A |
| Secondary voltage | 300 V (1st harm.) | 1.5 kV | 15 kV |
| Secondary current | $0 \div 10$ A (1st harm.) | 3 A | 40 mA |
| Output frequency | $10 \div 200$ Hz | - | 20 kHz |

The AC/DC network converter is a three-phase transistor bridge in 6T + 6D. It is a typical voltage inverter structure. An integral part of the AC/DC converter are $L_S$ line chokes connected in a series to the three-phase circuit input. The role of the AC/DC converter is to maintain a constant, preset $U_{dc}$ voltage at the DC output. It works as a boost converter. A boost of the AC/DC converter output voltage causes the DC/AC output inverter powered from the higher $U_{dc}$ voltage to form the first voltage harmonic of the higher amplitude at its output. This translates into a higher voltage supplying the plasma reactor. The AC/DC converter output has a parallel capacitive filter.

The DC/AC output converter is designed as a six-phase inverter system. Because the receiver, which is a plasma reactor, is of a non-linear character for the correct operation of the reactor and safe operation of the converter, the method of controlling the operation with a closed, non-linear current control loop (delta-modulation method) was adopted. The DC/AC converter has been designed as a six-phase inverter system in which the output voltages (currents) in individual phases are equal in amplitude and shifted by 60 (Figure 5A). The converter can also be considered as two independent three-phase inverters. Then, the output voltages (currents) have the same amplitudes for each three-phase system and the phase shift between them is 120. Two three-phase converters also allow for shaping output voltages (currents) with any phase shift between the two three-phase systems (Figure 5B).

The voltage at the AC/DC/AC converter output is too low to initiate ignition and hold the discharge in the plasma reactor. The voltage is adjusted to the plasma requirements using three single-phase transformers. Conventional voltage booster transformers with lower and upper voltage winding coaxially wound on one column are used. Single-phase transformers were used in order not to magnetically connect through the transformer's phase core, as would be the case if one three-phase

transformer was used. Such an approach eliminates the problem of interference transfer between phases. The parameters of the transformers are given in Table 2.

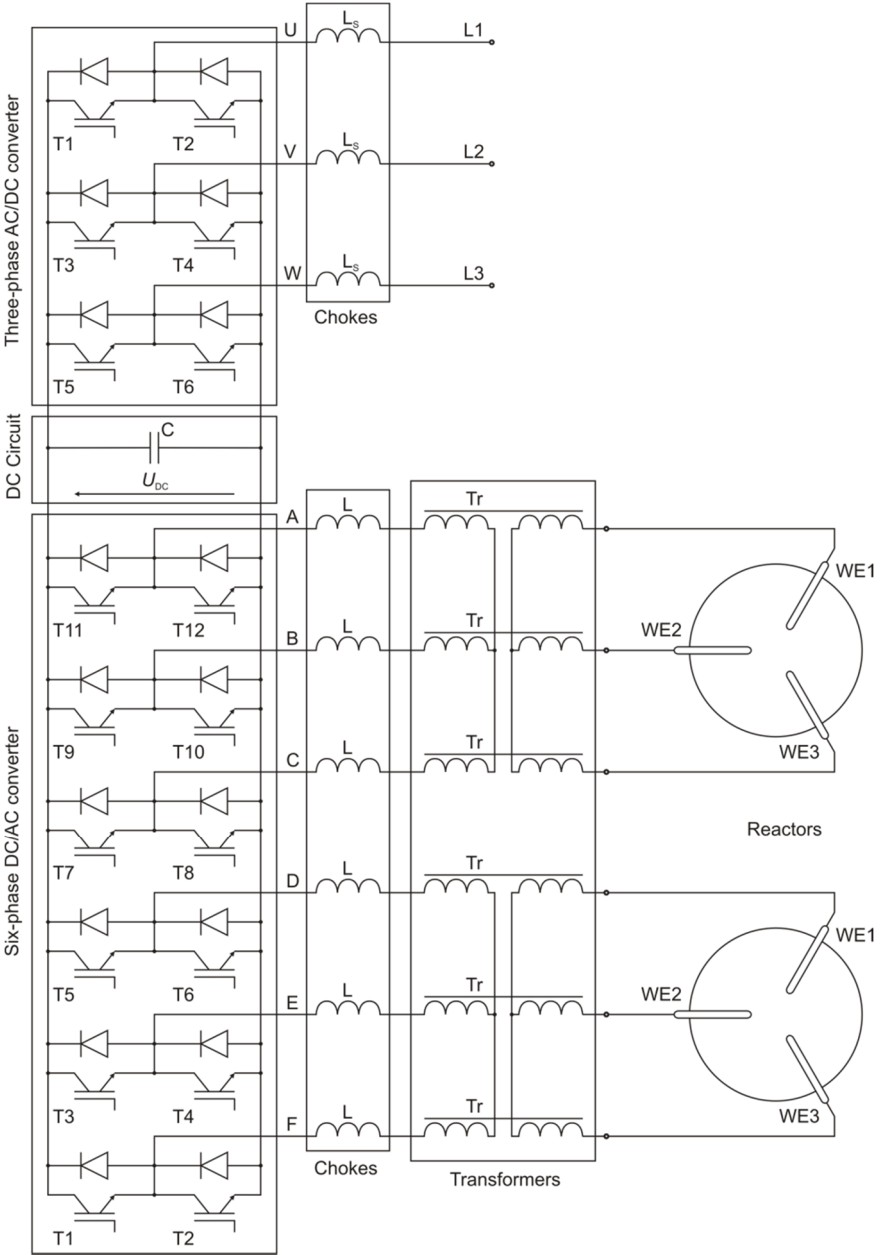

**Figure 4.** Plasma reactor power supply diagram by AC/DC/AC converter.

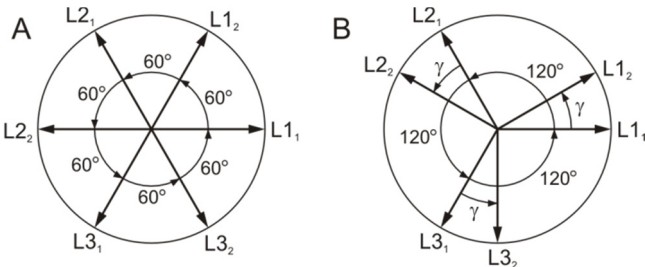

**Figure 5.** Six-phase converter output voltage vector system (**A**), or two three-phase converters with angle adjustment $\gamma$ (**B**).

Electronic voltage boosting modules were used to power the ignition electrodes of the plasma reactor. Such a module consists of a circuit of a high-frequency generator and a transformer that increases voltage [5]. The module's task is to generate low-power electric sparks. The parameters of the modules are given in Table 2.

## 5. Interaction of the Power Supply with the Reactor

Due to the low voltage of the secondary side of the AC/DC/AC converter (Table 2), it becomes necessary to increase the reactor supply voltage through matching transformers (Table 2). Tests of the converter-transformer-reactor system have shown that the characteristics of the arc plasma reactor receiver and the characteristics of the AC/DC/AC converter power supply system mean that the matching transformers with cores from conventional transformer sheets do not ensure the proper cooperation of both devices. This is manifested by the difficulty of obtaining the discharge power of the value that results from the power supply system used. Better cooperation conditions are provided by transformers with cores made of amorphous material Metglas 2605SA1. Using matching transformers with cores made of Metglas, higher discharge power is obtained at the same values of voltage, current and power frequency than in the case of transformers with cores made of traditional transformer sheets. This is due to the fact that Metglas carries a wider band of higher harmonics generated by the electrical discharge.

Matching transformers with Metglas cores significantly deteriorate the operating conditions of the AC/DC/AC converter. Generated by electrical discharge, higher harmonics, interferences and surges of up to several hundred kHz are transmitted through the transformers to the output of the converter, disturbing its operation.

The tested AC/DC/AC converter can operate as a source of voltage or current to supply the plasma reactor. When the converter is loaded as a source of current with a burning discharge in the plasma reactor, the voltage measured at its output is strongly deformed. FFT voltage analysis shows a high content of higher harmonics with a clear 10 kHz frequency advantage. The current retains its sinusoidal course. During the operation of the converter as a voltage source, a strong deformation of the arc current was observed with simultaneous slight deformation of the voltage course from the sinusoidal. Strong deformations of the current and discharge voltage disturb the operation of the converter. Since the DC/AC converter control system operates with a closed non-linear current control loop, it is necessary to use output chokes to limit the speed of output current changes. In order to limit the rate of current change in the receiver, which is a plasma reactor, it has become necessary to switch on 20 mH chokes at the output of the converter.

The long-term operation of the reactor has shown that the quality of the ignition system affects the correct operation of the plasma reactor, power supply system, electrode erosion and stability of the plasma process parameters. The reliable, repeatable ignition of the discharge guarantees the placement of the ignition electrode in the axis of symmetry of the system of working electrodes, in the lower part, in the place of the smallest inter-electrode gap (Figure 6A). This electrode is made of appropriately profiled tungsten wire with a diameter of 1 mm. The initiating spark discharge burns between the ignition electrode and the working electrodes. An important disadvantage of the solution is the high sensitivity to asymmetry of inter-electrode gaps, erosion of the electrodes and uneven flow of process gas, as a result of which, the initiating spark discharge channel closes with different intensity to the individual working electrodes. This results in an asymmetrical load on the output of the power supply system and a different degree of erosion of the working electrodes. Deepening with time, the uneven erosion of the electrodes translates into a further increase in the asymmetrical load on the power supply outputs. In the extreme case of asymmetry, one of the electrodes may stop participating in the operation of the reactor. The problem is the occurrence of high strongly deformed voltage, with a high content of higher harmonics and short-term overvoltages in the working electrode supply circuit, generated by the spark discharge (Figure 7A). This voltage is transferred to the primary side of transformers supplying working electrodes and further to the AC/DC/AC converter output.

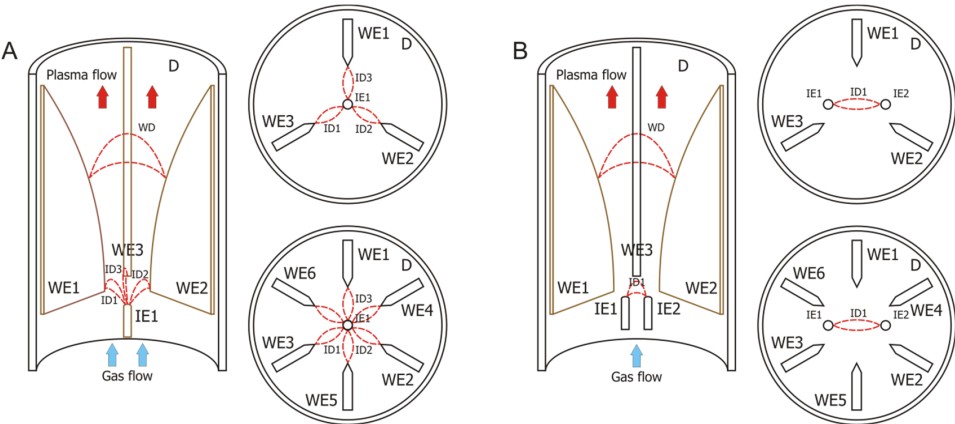

**Figure 6.** Reactor with one ignition electrode (**A**), and with two ignition electrodes (**B**).

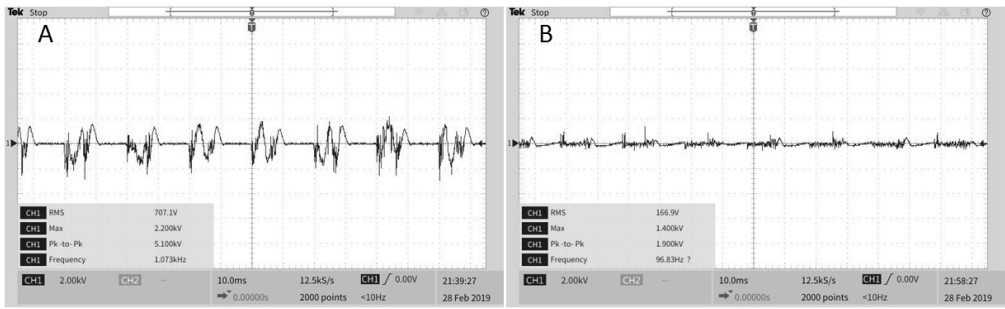

**Figure 7.** Voltage induced on the secondary windings of the transformer by an ignition discharge in still air for one ignition electrode (**A**), and two ignition electrodes (**B**).

Interference from the plasma reactor to the AC/DC/AC converter is transmitted to a lesser extent when the spark ignition spark discharge is ignited between the two ignition electrodes (Figures 6B and 7B). The absence of the ignition spark discharge to the working electrodes does not significantly affect the deterioration of the conditions for ignition of the main discharge. The absence of spark channels from the ignition electrode to the working electrodes, which are the beginning of the arc discharge, is compensated by a sufficiently high degree of ionisation of the space between the working electrodes. The ionisation must be chosen in such a way that, for a given process gas and its gas-dynamic parameters, the resistance of the gap between the working electrodes is reduced to a degree sufficient for spontaneous ignition of the arc discharge at reduced voltage. The appropriate degree of gas ionisation in the space between the working electrodes is obtained by selecting the parameters of the working electrode supply, i.e., voltage, current and frequency.

The voltages induced on the secondary side of matching transformers, as shown in Figure 7, are transferred to the output of the AC/DC/AC converter less the transformer gearbox value of 6.5. For both one and two ignition electrodes, these voltages do not exceed those which could damage the converter.

The design problem of power supply systems with AC/DC/AC converters is susceptibility to interference generated by electrical discharge. The discharge in a plasma reactor can be treated as a dynamic, non-linear conductivity with rapid, often random, time changes of currents, voltages and power, generating overvoltages, higher harmonics and electromagnetic interference, both conductive and radiated [6,7]. Observing the discharge parameters and analysing the level of current and voltage deformation and the level of conducted and radiated electromagnetic interferences emitted by the plasma reactor, it is concluded that appropriately selected transformer power supply systems provide better conditions for plasma generation, but without the possibility of smooth regulation of its parameters, which is a big disadvantage.

By improving the design of the reactor in terms of the selection of the shape of the electrodes, the conditions of plasma generation and the operation of the converter power system are improved. The shape of the electrodes, length between them [8] and their material influences the stability and burning time of the arc discharge. The electrodes are selected depending on the composition of the process gas, its gas-dynamic conditions and the parameters of the electricity supply system.

For nitrogen plasma, the electrodes used are convex, knife-edge, working edge profile, height 143 mm, width 30 mm (Figure 8A) and thickness 2 mm. The second stage was equipped with concave-shaped electrodes, 200 mm high, 30 mm wide and 2 mm thick (Figure 8B). The electrode material is 0H18N9 stainless steel. For this type of electrodes, the best burning conditions and the lowest interference of the AC/DC/AC converter with the adopted geometrical configuration of nozzles, gas-dynamic and electrical parameters were obtained.

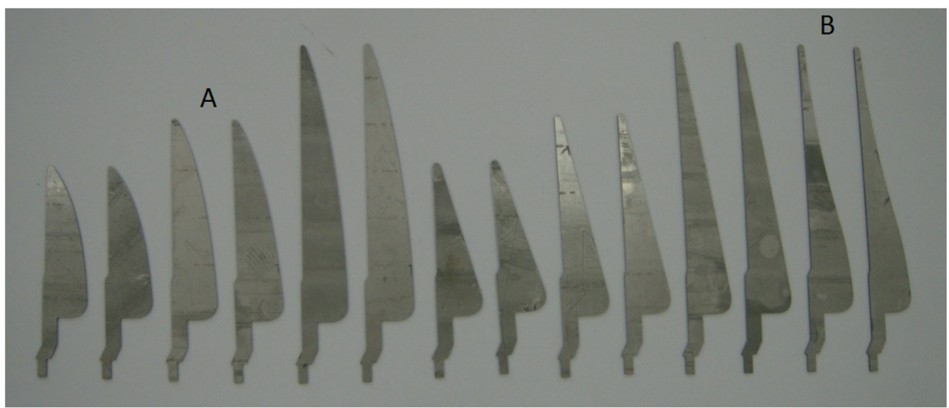

**Figure 8.** Example shapes of plasma reactor working electrodes.

## 6. Plasma Reactor Operation

The nature of the discharge in the reactor depends on the gas flow rate through the discharge chamber [9–11]. For small flows, the discharge is a typical arc discharge (Figure 9A). For large flows, the arc column is blown and the discharge has different characteristics from a typical arc discharge (Figure 9B). The discharge in the air for large gas flows, on visual observation, has the characteristics of a glow discharge (Figures 9A and 10A) and only pictures taken with a high-speed digital camera in the absence of light show the true nature of the discharge (Figure 10B).

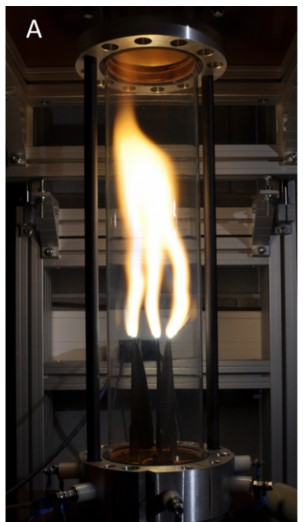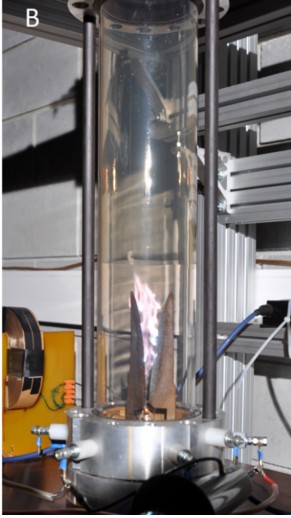

**Figure 9.** Air discharge for small gas flows (air, 0.5m$^3$/h) (**A**) and large gas flows (air, 10 m$^3$/h) (**B**); 30 fps camera, 1.5 kV on the secondary side of idle transformers.

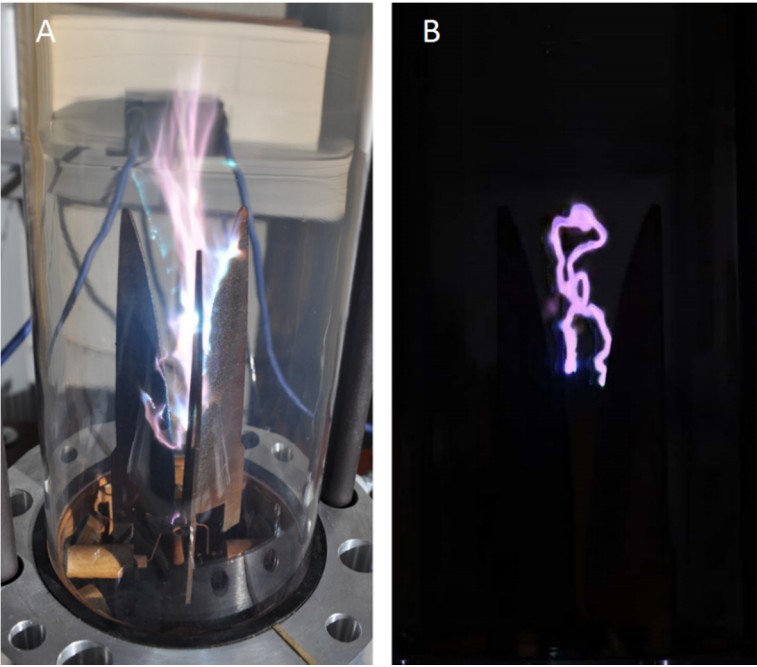

**Figure 10.** Air discharge for high gas flow rates (air, 10 m³/h) with 30 fps camera (**A**) and 10,000 fps high-speed camera (**B**); 1.5 kV on the secondary side of idle transformers.

While in parallel operation of plasma reactors, the electrical discharges occurring in them are of the same character, with fluctuations of some parameters within small limits, and in the cascade system, the discharges burning in particular degrees are clearly different (Figure 11).

The chemical composition of the working gas has a great influence on the operation of the reactor and parameters of the generated plasma [12]. Discharges obtained in the reactor differ significantly in character (Figures 12 and 13) depending on whether the plasma-generating gas is argon (Figures 12A and 13A), nitrogen (Figures 12B and 13B), helium (Figures 12C and 13C) or oxygen (Figures 12D and 13D).

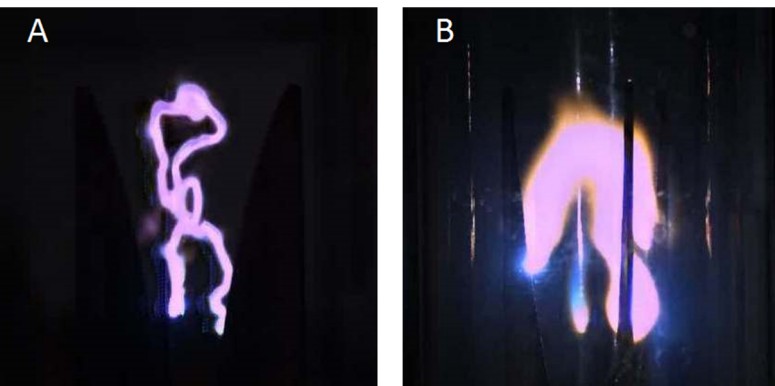

**Figure 11.** Air discharge in the lower (**A**) and upper (**B**) stages of the reactor cascade at high gas flow rate (air, 10 m³/h) with 10,000 fps high-speed camera, 1.5 kV.

The properties of the gas, and thus, the nature of the discharge in the reactor chamber change not only with changes in electricity supply parameters. Explaining the differences in the discharge is a complex problem even within the properties of one gas. Discharges for one gas differ in character depending on whether they burn in the first (Figure 12) or second (Figure 13) reactor stage. The gas in a reactor chamber often has a complex chemical composition. The individual particles have different properties. For example, the products of electrode erosion can be both single atoms and relatively

large pieces of material torn from electrodes. Gas contamination changes the electrical resistance to puncture of individual areas in the reactor chamber.

The obtained photographs illustrate how much, apart from the chemical composition and gas parameters, the character of the discharge is affected by the parameters of the reactor discharge chamber: decomposition of the reactor's structural elements, pressure distribution, distribution of flow rates, surface structure of structural materials, gas turbulence, temperature distribution, ionisation distribution, gas and discharge interactions, electromagnetic field distribution, impurities content and distribution, erosion of electrodes and their material and shape [13,14].

The electrical discharge also affects the gas flow to varying degrees, depending on the nature of the discharge. Only a small part of the gas enters the discharge and a large part of the discharge flows away. The discharge affects the gas dynamically, thermally and electromagnetically, changing the gas parameters in different areas of the reactor chamber to varying degrees.

The influence of gas on the nature of the discharge in the reactor is a complex and multi-threaded issue, difficult to describe and numerical simulations.

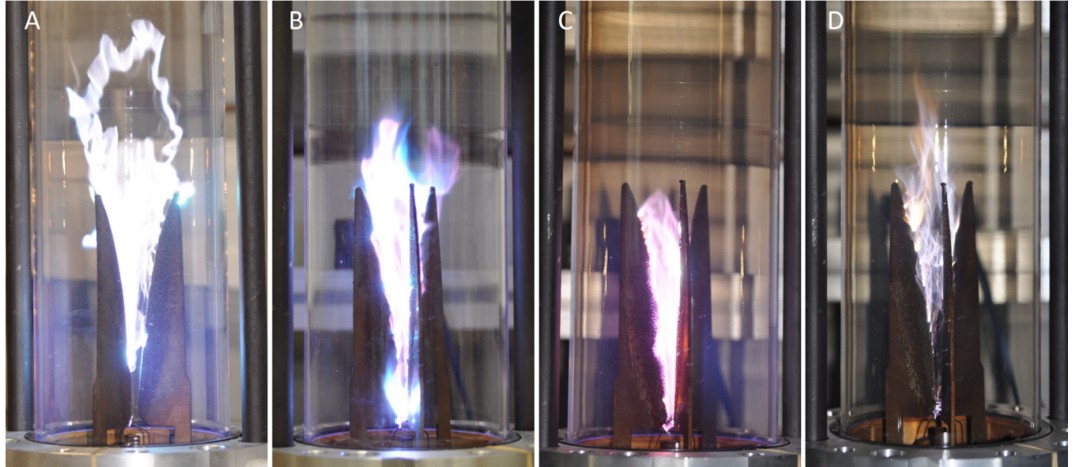

**Figure 12.** Discharge for argon (**A**), nitrogen (**B**), helium (**C**) and oxygen (**D**) in the first stage of a plasma reactor (10 m$^3$/h), 30 fps camera, 1.5 kV.

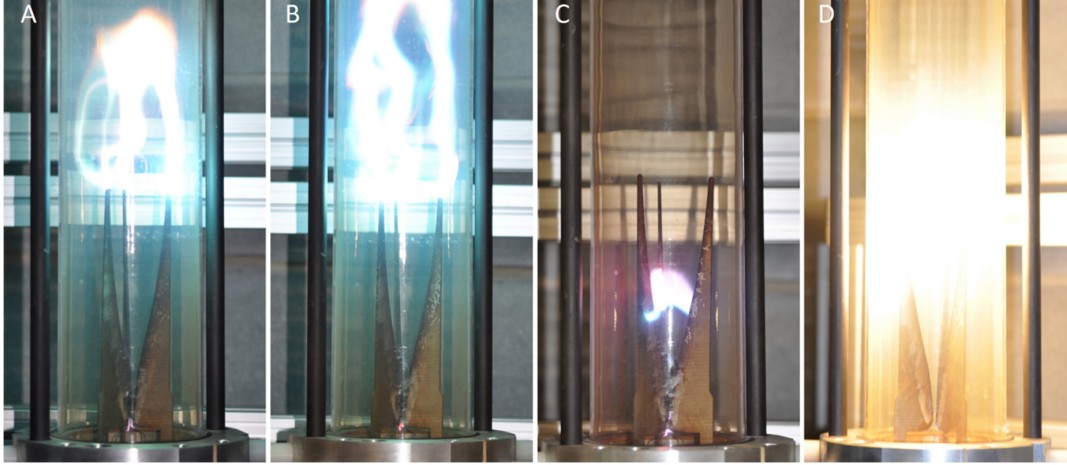

**Figure 13.** Discharge for argon (**A**), nitrogen (**B**), helium (**C**) and oxygen (**D**) in the second stage of a plasma reactor (10 m$^3$/h gas flow rate in the first stage of a plasma reactor), 30 fps, 1.5 kV.

Phase voltages, i.e., those measured on the reactor operating electrodes, for the gases mentioned above are shown in Figure 14 for the first reactor stage and Figure 15 for the second reactor stage.

The oscillograms were obtained for the nominal parameters of the reactor's power supply in electricity and process gases.

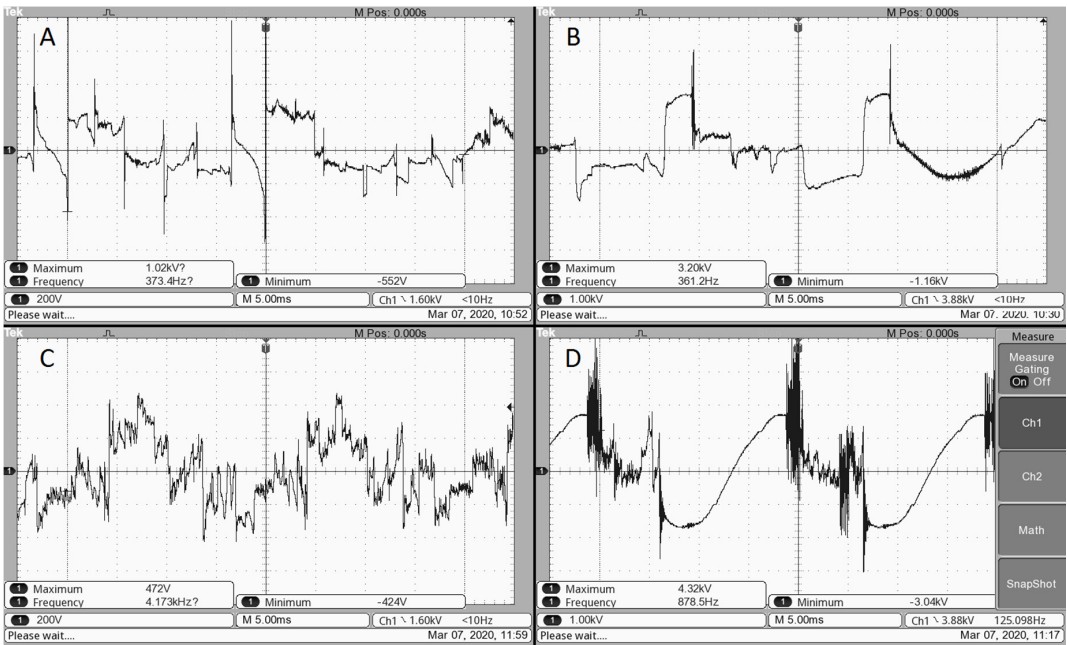

**Figure 14.** Voltage course on the reactor's first stage operating electrodes at nominal gas flow (5 m³/h) and nominal power supply parameters, argon (**A**), nitrogen (**B**), helium (**C**), and oxygen (**D**), 1.5 kV.

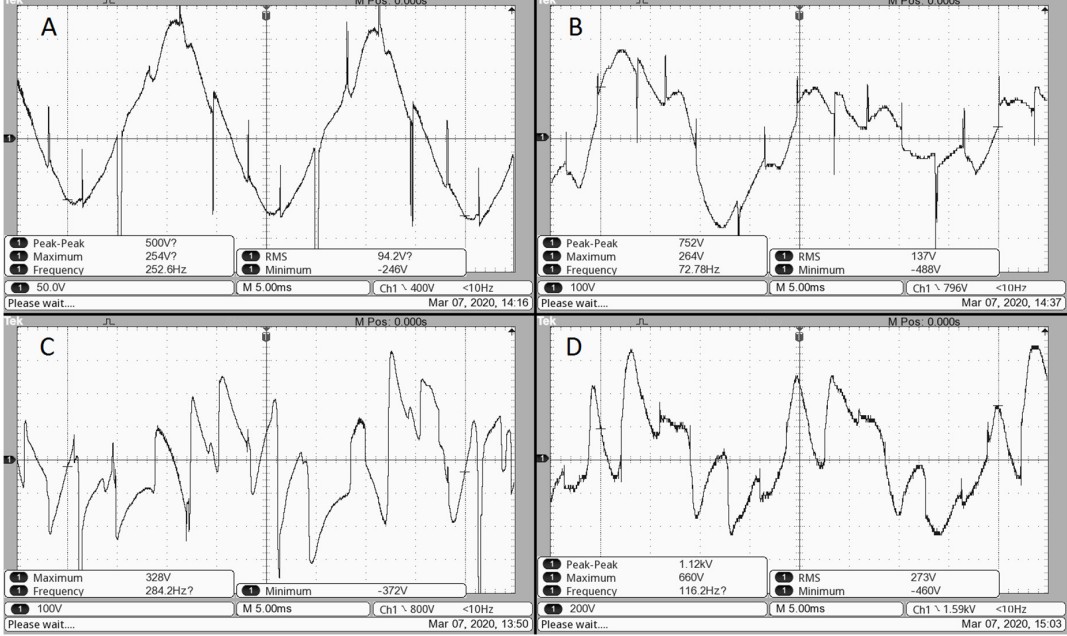

**Figure 15.** Voltage course on the operating electrodes of the second stage of the reactor at nominal gas flow (5 m³/h) and nominal electricity supply parameters, argon (**A**), nitrogen (**B**), helium (**C**), and oxygen (**D**), 1.5 kV.

Analysing the influence of the shape of the working electrodes on the operation of the plasma reactor, it is concluded that long knife-shaped electrodes (Figure 8) allow the obtaining of higher discharge capacities (4.8 kW for argon) with the same parameters of electricity supply and process gases as short electrodes (4 kW for argon). However, long electrodes generate more interference to the

converter power supply. In the case of long electrodes with a concave working edge, intermediate power (4.7 kW for argon) is obtained with by far the highest level of interference.

The operating problem of the plasma reactor is the pollution of the plasma-gas environment through soot, material from electrode erosion and the dust of other substances accumulating inside the discharge chamber. The high concentration of these materials changes the conditions for ignition and development of electrical discharge. Pollutants enter into chemical reactions with substances subjected to plasma treatment, cause changes in energy parameters, disrupt the continuity of technological processes and increase the operating costs of the plasma reactor.

The deposition of conductive dusts such as coal or metals on the reactor components often leads to short circuits of the electrodes due to a reduction in the surface resistance of the insulating components of the reactor. Short-circuiting of the ignition electrode to the working electrodes can be particularly dangerous for AC/DC/AC converters. In this case, an ignition voltage of 15 kV with a frequency of 20 kHz is transferred to the output of the converter, minus the gear of the matching transformer to 2.3 kV.

Sources of contamination other than process gases are products of reactor electrode erosion. Non-useable reactor electrodes should have high melting point, low concentration of contained gases and redundant, easily evaporating admixtures, high chemical resistance and surface structure stability, low output operation and high speed of electrode stain movement. Apart from physical factors, such as cooling capacity and chemical composition of the gas, a significant influence on electrode erosion is exerted by electrical parameters of the discharge determined by the power supply system.

## 7. Summary

The interaction of an AC/DC/AC converter with an arc plasma reactor is a multithreaded and complex issue, difficult to analyse due to mutual correlations of a few parameters in many devices.

The GlidArc Plasma Reactor is sensitive to the changes of many parameters, such as electrical (voltage, current and frequency); gas-chemical (chemical composition of process gas, plasma treatment products, humidity); gas-dynamic (gas pressure and flow); mechanical (electrode erosion). Thus, the arc plasma reactor is a device, which is difficult to diagnose in real time both from the point of view of plasma parameters and from the point of view of control of its technical performances. The difficulties are aggravated by non-linear characteristics of the reactor and a large number of different types of interference, often of a random nature.

When selecting the operating parameters of a plasma reactor and power and process gas supply systems for the requirements of the process, it is necessary to determine which of the parameters must be maintained in strictly defined regimes and which may be out of control. The issue of plasma parameters is as complicated as we go deeply into its physical and chemical properties. The shaping of plasma parameters is not only limited to supply voltage, current and frequency. Many factors have to be taken into account, some of which are discussed in the article, such as: gas properties, distribution of flows in the reactor, share of impurities, shape and distribution of electrodes, distribution of electromagnetic field and interference. Many of these factors are difficult to measure and control, and some are in opposition.

The AC/DC/AC converter turned out to be susceptible to interference generated by the plasma reactor, and from this point of view, transformers are better suited as power supplies for plasma reactors. However, the converter power supply has an advantage over them when it is necessary to adjust the parameters of the generated plasma in real time. A disadvantage of the analysed power supply is the lack of possibility to adjust different settings of current, voltage and frequency values for both reactor stages.

It is noted that within certain limits of electricity supply and gas parameters, the reactor generates a greater number of interferences. However, it is difficult to systematise the influence of particular parameters on AC/DC/AC converter operation.

In a plasma reactor, it is difficult to maintain the vast majority of parameters at one strict level. In this case, a big problem was to ensure proper cooperation of the power supply with the plasma reactor. Interference generated by the reactor significantly deteriorated the operating conditions of the AC/DC/AC converter. Improvement of the cooperation of both devices was achieved by introducing many modifications of the reactor in relation to its original version. Two ignition electrodes were used instead of one, which significantly reduced the number of overvoltages dangerous for the power supply. It is possible that further improvement will be achieved by modifying the shape and position of the ignition electrodes. The shape and material of the working electrodes were chosen. By selecting the shape of the working electrodes, you can influence the stability of the discharge, and thus, reduce the number of interferences. The electrodes also function as antennas emitting interference. By selecting the material of the electrodes, you can reduce the content of erosion products in the reactor chamber, and thus, eliminate the danger of electrode penetration and short circuits. The article indicates the differences in discharge for different gases. By using instead one gas of their mixture in appropriate proportions, it is possible to improve the conditions for burning of the discharge, and thus, reduce the interference. Finally, there are certain values of voltages, currents and frequencies, at which the cooperation of the reactor with the power supply is not beneficial. The voltage must not be lower than a certain value typical of the gas and reactor design. At too low values, the discharge burns unstable, generating a lot of interference. In the case of the current, higher values increase the discharge temperature, which translates into fewer surges and interferences. Finally, by optimizing the construction of the reactor in terms of gas-dynamics, it is possible to achieve a reduction in interference generated by the discharge.

Matching transformers play an important role in the cooperation between the converter and the plasma reactor. By shaping their characteristics in terms of the size of the winding leakage reactance and the material properties of the core, optimum conditions of cooperation between the plasma reactor and the converter can be ensured.

**Author Contributions:** These authors contributed equally to this work. G.K. and M.A. writing—original draft preparation, G.K. and M.A. writing—review and editing, G.K. and M.A. photos and graphic design, G.K. and M.A. test bench design and measurements, G.K. and M.A. the concept of modifications and their implementation, G.K. and M.A. analysis and processing of measurement data. All authors have read and agreed to the published version of the manuscript.

**Funding:** This research did not receive any specific grant from funding agencies.

**Conflicts of Interest:** The authors declare no conflict of interest.

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
