# Peer review of "Operating Problems of Arc Plasma Reactors Powered by AC/DC/AC Converters"

_applsci, doi:10.3390/app10093295_

Round 1

Reviewer 1 Report

The paper explores gliding arc plasma reactors (GlidArc) used to convert electrical energy on specific electrical discharges. These reactors could provide free radicals under nearly atmospheric pressure conditions for some industrial applications. In particular the proposed AC/DC/AC converter is used to regulate the voltage, current and frequency by compensating unwanted fluctuations in the parameters of the plasma process regardless of process gas composition, pressure, and chemical purity. A modular design, in parallel or in series is also proposed with different number and geometry of electrodes at given power supply parameters. The paper is rich in information nevertheless no real guidelines are proposed to overcome the problems. I will suggest enriching this part by providing a concreate example.

  • All operating parameters in Figures 9-14 must be reminded in the legend.
  • There are two Figures 11.
  • Due to the non-linear behavior and the huge number of parameters of the power supply system (Table 2), computer simulation could be very helpful to correlate voltage, current and power outputs with the gliding electrical discharge. Please comment.
  • What is the effect of the electrode geometry showed in Figure 8 on consumed energy in the reactor as a function of applied voltage?
  • One of the missing information from the paper is the correlation between the electrical parameters of the discharge and plasma parameters. This could be down through reviewing the pertinent scientific literature in the introduction and/or providing some additional data from the authors.
  • The authors did not well explain how to regulate the voltage, current and frequency to compensate unwanted fluctuations which is the key concept of this manuscript. I will suggest enriching this part of the manuscript by a concreate example and practical guidelines.

Reviewer 2 Report

The paper is well written, with a detailed description of the experimental apparatus and a clear description of results. However, I feel it lacks in highlighting what the novel aspects are and what the scientific impact of these  is. Other comments are below:

  • Page 8, line 237. The authors should explain why the discharge is different for different flowrates
  • The authors should explain why different gas composition generate different discharge types.
  • Page 10, lines 258-260. This needs more details and explanation, not just at qualitative level
  • The operating problem of the plasma reactor are mentioned but it is not clear if these were analysed in this work and what advantages the new power supply brings about

Round 2

Reviewer 1 Report

The authors adressed most of the unclear points and provided a better version of the paper.

Mathematical description of the gliding arc discharge even a complex and highly dynamic problem was already treated in the scientific literature via  "black box" electric arc models [1] to more sophisticated models including plasma chemistries in 2D and 3D [2-4] including CO2 conversion among several other papers.

[1] L. JaroszyÅ„ski, H. D. Stryczewska, “THE NUMERICAL MODEL OF THE GLIDING ARC DISCHARGE”, Conference: II International Symposium: New Electrical and Electronic Technologies and Their Industrial Implementation NEET’2001

[2] St Kolev and A Bogaerts, A 2D model for a gliding arc discharge, Plasma Sources Science and Technology, Volume 24, Number 1, 2015

[3] St Kolev and A Bogaerts, “Three-dimensional modeling of energy transport in a gliding arc discharge in argon” Plasma Sources Science and Technology, Volume 27, Number 12, 2018

[4] A 2D model of a gliding arc discharge for CO2 conversion, AIP Conference Proceedings 2075, 060008 (2019); https://doi.org/10.1063/1.5091186

Reviewer 2 Report

The authors have provided a good response to my queries and modified the manuscript where required.